# Combined Discharge and Thermo-Salinity Measurements for the Characterization of a Karst Spring System in Southern Italy

**Ivan Portoghese** [1] , **Rita Masciale** [1] , **Maria C. Caputo** [1] , **Lorenzo De Carlo** [1] and **Daniela Malcangio** [2,*]

[1]    Water Research Institute, National Research Council of Italy, 70132 Bari, Italy; ivan.portoghese@cnr.it (I.P.); rita.masciale@cnr.it (R.M.); maria.caputo@ba.irsa.cnr.it (M.C.C.); lorenzo.decarlo@cnr.it (L.D.C.)

[2]    DICATECH, Polytechnic University of Bari, 70125 Bari, Italy

*    Correspondence: daniela.malcangio@poliba.it

**Abstract:** The hydrological monitoring of springs is an auxiliary and indispensable tool that goes alongside investigations in wells to reconstruct a conceptual phenomenological model of an aquifer–groundwater system and its interactions with surface waters. There are manifold ways to carry out this monitoring, but the choice of which way is significant for a correct qualitative and quantitative knowledge of spring systems. The present work focuses on the characterization of the thermo-saline and flow regimes of the Tara spring system along the northern coast of Taranto (southern Italy), where a karst groundwater basin is the major source of the Tara River and the surrounding coastal wetland. A series of measurements was carried out on the spring system to support a technical feasibility study on the possible use of the brackish water of this river to feed a future desalination plant. To estimate the flow rate, a comparison was made between different flow measurement methods in a derivation channel. Through an analysis of the available dataset, the response of the aquifer to the autumn–winter recharge, for which updated hydrologic measurements were not available, is highlighted.

**Keywords:** Tara spring system; electrical conductivity; discharge measurement; flow measurements; current meter method

## 1. Introduction

As in many other Mediterranean regions, the economy of southern Italy is mainly based on irrigated agriculture. As a result, water resources have seen a serious qualitative and quantitative depletion caused by overexploitation. Due to the lack of significant surface watercourses, there has been a greater dependence on groundwater, and a number of irrigation wells have been drilled to provide the necessary water. In some groundwater basins of southern Italy, irrigation channels draining groundwater flows are an alternative source, providing high-quality water to croplands and regular discharge regimes.

In more recent years, policies related to freshwater conservation are likely to become obsolete or no longer responsive to new and changing environmental and socioeconomic conditions [1]. It has been observed that climate change, combined with anthropogenic pressure and infrastructure, is consistently associated with changes in several components of the water systems [2], particularly concerning the Mediterranean region [3].

Springs, in particular, are very vulnerable to long-term climate alterations, land surface activities, intensive agriculture, growing water demands, and landscape alterations [4,5]. Therefore, there is a need to update studies on large groundwater spring sheds and investigate the impact of climate

change on the potential groundwater recharge [6–8]. The monitoring of springs involves the study of its physical, chemical, and biological phenomena, using appropriate equipment with the aid of analytical techniques. Moreover, spring monitoring is an indispensable instrument (after the investigation of wells) in reconstructing reliable conceptual models of groundwater–aquifer systems [9]. A qualitative and quantitative description of groundwater spring processes can only be achieved through a thorough understanding of relationships between the geological, hydrogeological, and chemical–physical elements that characterize the aquifer system, as well as their evolution over time. In this regard, special attention has recently been addressed to the principal karst aquifer systems in central and southern Italy due to observed climate-induced alterations in the drinking water supply (supplying about 12 million people) [10].

Among the existing investigation techniques, water tracing methods have been carried out over the years to define catchment boundaries, estimate groundwater flow velocities, determine areas of recharge, and identify sources of pollution of spring water [9]. Luhmann et al. [11] presented an overview of the different methods used for the study of spring behavior within a karst aquifer as a means of characterizing a groundwater basin from spring characteristics. They outlined the hydraulic [12], chemical [13], and isotopic [14] responses from spring monitoring to provide information about the aquifer. Additionally, a combination of temperatures and heads was jointly suggested in order to provide unique information on springs. Indeed, Luhmann et al. [11] asserted that karst spring temperature is the most crucial source of information for determining aquifer characteristics. Alexander et al. [6] examined a further methodology for a better understanding of the overall flow system in southeastern Minnesota, using a combination of dye tracing data, the water well driller's records, and downhole gamma logging. Furthermore, electrical tracing methods based on geophysical techniques are also highly promising in investigating surface water–groundwater interactions, particularly on hillslopes [15]. Both temperature and the electrical conductivity of spring water have been observed to be closely related to spring discharge [16,17]. For example, Birk et al. [18] showed that localized spring discharge in a karst aquifer could be characterized by simultaneously analyzing the electrical conductivity and temperature of spring water.

Broadly speaking, chemical–physical parameters, which are recognized as typical "natural tracers", are closely related to the lithological characteristics of a rock mass, to the type of water circulation in an aquifer, and to the type of recharge, which can be autogenic or allogeneic [19–21]. In this type of monitoring, the absolute values are not as important as the variations that these tracers may undergo over time as a result of the increase in flow rate. In particular, the temperature of spring water is an easy parameter to measure continuously. Furthermore, the interactions between rock and water are governed by relatively simple laws such as heat exchange, which is linked to water–rock contact and residence times. Heat exchanges between rock (characterized by low thermal conductivity) and water (high thermal conductivity) affect the temperature at the source. The specific electrical conductivity is relatively more difficult to measure continuously and is linked to the total mineralization of the water. It is a physical parameter that depends on the content of the main dissolved ions (essentially bicarbonates, calcium, magnesium, sodium, potassium, chloride, and sulphate) and therefore indirectly reflects the chemical load of the groundwater. It is therefore the concentrations of these ions and their variations over time that determine the trend in the electrical conductivity at the source and therefore the degree of mixing between waters of different origins.

The present research sets the stage for a technical and environmental feasibility study about the construction of a desalination plant by the local water utility using waters emerging from springs. A three-year monitoring program was implemented to determine the quantity (i.e., the flow rate) and quality (i.e., the temperature and conductivity) of the spring system, called Tara, for which updated hydrological measurements were not available. The results were then used to estimate the feasibility of the project, as well as to provide important information on the status of the aquifer. While the specific results of this study are of local interest, the approach of using a combination of (i) established methods to estimate the channel flow rate and the residual capacity of the karst spring system and (ii) physical

parameters to determine the qualitative characterization of the spring system may also be considered relevant. This applies to the entire scientific community dealing with the evaluation, development, and management of groundwater resources, particularly in less developed regions, where hydrological monitoring is often discontinuous or unavailable.

## 2. Study Area

From a geological point of view, the study area is located in the SW sector of the Murge relief, which corresponds to the southern Apennines foreland (Figure 1). From the bottom to the top, the following geological units can be distinguished [22]: Altamura limestone (Cretaceous), Gravina calcarenite (Upper Pliocene–Lower Pleistocene), Subappennine clays (Lower Pleistocene), terraced marine deposits (Middle–Upper Pleistocene), and alluvial and coastal deposits (Holocene). Several normal faults dislocate the Cretaceous bedrock, creating a horst and graben setting oriented NW–SE. As a consequence, a Plio–Quaternary deposit outcrop on the lower structural sectors of the area conceals faulted Cretaceous limestone, which, conversely, is found only on the highest structural sectors (Figure 1a). This geological setting also includes two hydrogeological complexes indicative of specific aquifer structures: a deep limestone aquifer (the main aquifer) characterized by secondary permeability caused by fracturing and karstic processes and a shallow porous aquifer corresponding to a sand–conglomerate–calcarenite complex of terraced marine deposits and alluvial–coastal successions [23].

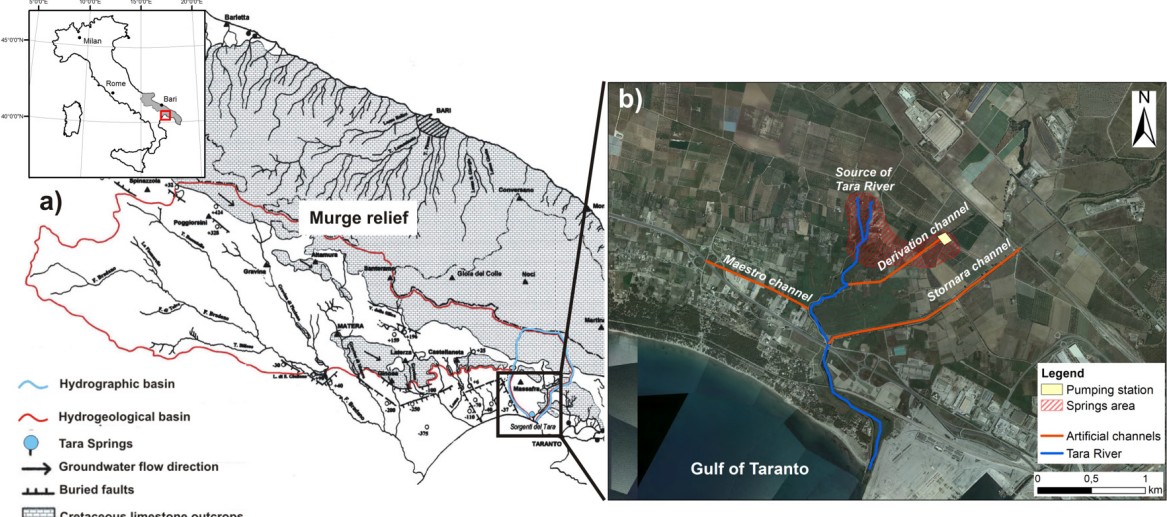

**Figure 1.** Study area: (**a**) extension of the Tara's hydrogeological basin; (**b**) indication of the spring area and the water system of the Tara River.

The Tara spring system represents the most important karst spring in the Apulia region fed by groundwater circulating within the deep limestone aquifer, which originates from the Murge relief.

The presence of the impermeable covering deposits (Subappennine Clays), going toward the sea, makes the aquifer confined and locally even artesian, as confirmed by several flowing artesian wells in this area. Moreover, the deepening of the clay deposits hinders the seaward water flow and thus forces it to flow upwards along higher permeability paths controlled by structural discontinuities in the bedrock or by permeability changes in the underlying deposits. Consequently, a large water-rising zone results, located in the NW sector of Taranto (Figure 1b). It consists of a main spring area about 2 km from the sea that gives rise to the Tara River and to several secondary springs (about 20) spread over a larger area. All springs form a single spring system with moderately saline water (~3 g l$^{-1}$). Studies dealing with the chemical characteristics of brackish groundwater sampled in different coastal springs of the Apulian karstic aquifers have revealed that the salinity of the Tara spring system is linked to aged saltwater, which is very different from modern seawater [24,25].

The Tara spring system generates regular outflows with peaks of about $4 \text{ m}^3 \text{ s}^{-1}$, according to measurements made in the first half of the last century. The remarkable outflows of the Tara spring system are fed by a hydrogeological basin that is much more extended than its hydrographic one (Figure 1a). Moreover, considering the limestone outcroppings, it is possible to identify a feeding area close to the hydrographic catchment of the Tara River and a farther one connected to the inland limestone outcroppings [23].

The Tara River, which flows into the Gulf of Taranto after a short stretch of about 3.5 km (Figure 1), is almost exclusively fed by deep groundwater input, as evidenced by the high electrical conductivity values of its water and an almost constant temperature of 18–20 °C throughout the year. In the early 1950s, a capture plant for one of the secondary springs was built about 1.2 km from the source of the Tara River by constructing a pumping station consisting of an intake chamber carved out of limestone and equipped with a pumping system. In addition, an artificial channel 3 m wide, 3.5 m deep, and 1000 m long was built as a derivation channel connecting the Tara River to the pumping station. The side walls of this channel, which have a concrete lining, prevent any lateral inflow and outflow, thus allowing the channel to drain any groundwater outflow from the channel bottom. Indeed, the function of the channel is to intercept additional sources and collect them toward the pumping station. The latter consists of eight different electric pumps with a total power of 750 kW that are able to operate at a total flow rate in the range of $0.40$–$3.70 \text{ m}^3 \text{ s}^{-1}$. A movable dam was built on the natural stream downstream of the confluence with the derivation channel to enable a more efficient and flexible operation of the pumping station. In this way, water can flow in either direction through the derivation channel, i.e., from the intake chamber to the confluence with the river and vice versa, depending on the hydraulic gradient between the water heads in the river and the intake chamber.

From its construction until the early 1990s, the pumping station was operated at around full capacity, mainly supplying water for industrial needs linked to steel production and subordinately for agricultural use. However, at the present time, only one or two electric pumps are simultaneously active, so that water is withdrawn almost completely from the spring, which is intercepted just below the pumping station. Nowadays, the extracted flow rate ranges between $0.45 \text{ m}^3 \text{ s}^{-1}$ and $0.65 \text{ m}^3 \text{ s}^{-1}$ due to a drastic reduction in blast furnace activity in the Taranto industrial zone. Under such an exploitation regime, the flow direction in the derivation channel occurs from the pumping station to the natural stream. Consequently, a renewed interest in the hydrogeochemical characterization of the Tara spring system is justified both for the purposes of wetland conservation as well as for its potential exploitation as feed water for a water desalination plant.

## 3. Methods

The survey method adopted for the present study aimed to (i) detect the main springs intercepted along the derivation channel by means of thermoconductivity measurements; (ii) identify the optimal measurement method as a tradeoff between simplicity of execution and reliable estimates, taking into account the characteristics of the channel (narrow and deep) and the observed flow regimes; (iii) study the variability of the flow regime during a hydrological year through an appropriate measurement program; and (iv) evaluate the medium–long-term variations in streamflow regimes, also as a result of the decreased spring withdrawal.

### 3.1. Measurement Techniques

Natural traces provide valuable information about how groundwater moves in the subsurface and the characteristics of karst springsheds. Electrical conductivity (EC) can be used as an index of total dissolved solids and, in some cases, as a predictor of concentrations of individual ions. EC can also be used to interpret the changing sources of stream runoff at different time scales and to provide information about the contrasting hydrological behavior of specific catchments [26]. Because groundwater commonly differs chemically from stream water, groundwater discharge zones often coincide with relatively sudden changes in water chemistry along a stream, which can be detected

by measuring along-stream variations in EC. Inferences regarding groundwater discharge can be made more confidently by combining EC measurements with other observations, such as hydraulic gradients across the streambed, water temperature, and streamflow measurements. Furthermore, EC can be used to compute the relative contributions of two tributaries to flow below the confluence, or to separate quantitatively the contributions to streamflow from two distinct sources. As long as the limitations of EC are borne in mind, its measurement can provide useful and rapid insight into the chemical and hydrological characteristics of water systems. EC is relatively easy to measure either manually, using a handheld conductivity probe, or almost continuously using a probe connected to a data logger. EC measurements are strongly temperature-dependent, so they must be standardized to a reference temperature, typically 25 °C or 20 °C, to make the measurements comparable to each other.

In a recent review of methods used for streamflow monitoring, Dobriyal et al. [27] wrote that streamflow monitoring methods are specific to stream types, which can be classified on the basis of eight major variables. These are their width, depth, velocity, discharge, slope, roughness of bed and bank materials, sediment load, and sediment size. The different methods available to quantify and monitor surface water flow are grouped into four categories [28–30]: (a) direct measurement methods, (b) velocity–area methods, (c) formed constriction or constricted flow methods, and (d) noncontact measurement methods. A brief account of the suitability of each method to different terrains, with their advantages and disadvantages, is summarized in Dobriyal et al. [27]. Moreover, the selection of measurement method should be based on the volume of water to be measured, the degree of accuracy desired, whether the installation is permanent or temporary, and the required costs [31,32].

Considering that the investigated channel has regular, artificial walls, a width of 3 m, a natural stony–sandy bed, and a water depth of around 2 m, certain measurement methods were chosen and tested on multiple sections, which had been previously selected according to the thermo-salinity anomalies detected along the stream current. Based on the channel characteristics, two measurement methods belonging to the velocity–area category were adopted, namely the current meters method (CM method) and the float method (F method). Though they are very different in terms of complexity of operation and the accuracy of the measured discharge, the two methods were tested and compared, and we highlight the pros and cons in the given field conditions.

### 3.1.1. CM Method

The CM method is considered to be the most accurate of the velocity–area methods when appropriate procedures are adopted both for the measurement devices and field acquisition and processing. This method consists of determining a flow velocity in a cross-section of a stream or channel by means of a current meter and computing the discharge using known geometric relationships [33].

We adopted the midsection method for computing discharge, assuming that the mean velocity in each vertical represented the mean velocity in a partial section (segment). The mean velocity in each vertical was achieved from velocity observations at several points in that vertical and by using a known relation between those velocities and the mean in the vertical. As suggested by most common methods, velocity measurements were taken at 60% of the water depth, as we assumed this velocity to be representative of the average velocity along the given vertical. The sum of the discharges for all the partial sections was assumed as the total discharge of the stream section. Moreover, the vertical–velocity curve method was adopted in some cases where detailed current measurements were required to investigate the effect of submerged weeds on the velocity profiles and on the two-point mean velocity estimated values. In these cases, a series of velocity measurements was carried out at each of the verticals, i.e., at 0.1-m depth increases between 0.1 and 0.9 m of depth. The results obtained using the vertical–velocity curve method were compared to the two-point method for a mean velocity measurement.

A drawback of indirect methods is the presence of flexible (elastic) vegetation in the channel, which can lead to errors in velocity measures, especially when using reels. In fact, the submerged vegetation that flexes due to the dragging action exerted by water flow offers additional resistance to

motion and therefore affects flow velocity. Moreover, vegetation roughness is not constant but depends on the flow condition (depth and velocity) as well as the vegetation condition (type and density). According to the theory of Kouwen et al. [34], in the presence of flexible vegetation, the velocity profile exhibits a logarithmic trend, described as follows:

$$\frac{V_m}{u^*} = C_0 + C_1 ln\left(\frac{A}{A_v}\right),$$ (1)

where $V_m$ is the mean flow velocity; $u^*$ is the shear velocity, defined as $(gRJ)^{0.5}$, with $g$ being gravitational acceleration, $R$ being the hydraulic radius, and $J$ being the energy gradient (i.e., the hydraulic slope, $i$); $A$ is the area of the channel cross-section and $A_v$ the area of its vegetated part; $C_0$ is a parameter based on the vegetation density; and $C_1$ is dependent on the vegetation stiffness. The parameters $C_0$ and $C_1$ are tabulated (Table 1) based on the ratio between $u^*$ and the critical shear velocity ($u^*_c$), which is introduced to describe the limiting friction velocity between the erect and prone states of vegetation. Indeed, vegetation can be regarded as prone if $u^*$ is higher than $u^*_c$.

**Table 1.** Values of $C_0$ and $C_1$ in Equation (1).

| Configuration | Criterion | $C_0$ | $C_1$ |
|---|---|---|---|
| Erect | $u^*/u^*_c \leq 1$ | 0.42 | 5.23 |
|  | $1 < u^*/u^*_c \leq 1.5$ | 0.57 | 7.64 |
| Prone | $1.5 < u^*/u^*_c \leq 2.5$ | 0.79 | 8.71 |
|  | $2.5 < u^*/u^*_c$ | 0.82 | 9.9 |

To estimate $u^*_c$ and then the roughness deflection for flow over submerged flexible vegetation, Kouwen and Unny [35] introduced a stiffness parameter, *MEI*, which takes into account the density $M$ and the rigidity *EI* (elasticity, $E$, and the moment of inertia, $I$) of the vegetation and is defined as follows:

$$MEI = \gamma di\left[3.4h_d\left(\frac{h_v}{h_d}\right)^{0.63}\right]^4,$$ (2)

where $\gamma$ is the specific weight of water, d the average depth of the water, $h_d$ the deflected roughness height of the vegetation, $h_v$ the height of vegetation, and $h_v/h_d$ the inflection degree of vegetation. Therefore, $u^*_c$ is defined by two different expressions, depending on the deformation behaviors of the vegetation:

$$u^*_c = 0.028 + 6.33(MEI)^2,$$ (3)

$$u^*_c = 0.23(MEI)^{0.106},$$ (4)

where the first equation is valid in the case of flexible vegetation, and the second one is valid for vegetation that breaks when flattened. Based on the value assumed by the ratio $u^*/u^*_c$, the values of parameters $C_0$ and $C_1$ can be chosen (Table 1) and the velocity calculated. The flow rate is then calculated using the Chezy formula:

$$Q = \chi A \sqrt{R i},$$ (5)

where $\chi$ is the roughness coefficient assessed considering Manning's formula,

$$\chi = \frac{1}{n} R^{1/6},$$ (6)

in which $n$ nominally quantifies channel roughness or resistance to flow. In the case of vegetated river beds, $n$ can be calculated as follows:

$$n = 0.113R^{1/6}\left[1.16 + 2Log_{10}\left(\frac{R}{h_v}\right)\right]^{-1}.$$ (7)

### 3.1.2. F Method

Floats have somewhat limited use for stream gauging, but they prove useful when the velocity is too low to obtain reliable measurements with a current meter [33].

Similarly to the CM method, the F method is based on the velocity–area principle; therefore, the section geometry must be defined by measuring the depth and width of n subsections. A single float near the middle of the channel is used to determine surface velocity ($V_s$) as the averaged value of several measurements [36]. The technical literature reports that a surface float travels with a velocity about 1.2 times the mean velocity of the water column beneath it [37]. The velocity of any float (e.g., a wooden stick, a tube with a weight at its lower end), whether on the surface or submerged, is likely affected by wind, though with different impacts on measurement errors. The $V_s$ data collected from the measurements were processed by applying three different corrective methods:

- A quadratic relation obtained from the combination of the "Indian Standard for Velocity Area Methods for Measurement of Flow of Water in Open Channels (IS 1192–1981, Section. 5.3.2.3)" and the "Guide to Hydrological Practices" of the World Meteorological Organization (Volume 1, Table I.5.2) [38]:

$$F = -0.1333 \cdot \left(\frac{1}{d}\right)^2 + 0.18 \cdot \left(\frac{1}{d}\right) + 0.8433, \tag{8}$$

  where $d$ is the average depth of water in the upstream and downstream sections, and l is the sinking of the float below the free surface;
- The 1963 Roche study [39], which indicated that the appropriate $F$ for this case study was equal to 0.85 for all of the sections; and
- An empirical expression taken from the Mysore Research Institute in India [40]:

$$V_m = 0.8529 \cdot V_s + 0.0085. \tag{9}$$

Finally, the transit flow rate ($Q$) between the upstream and downstream sections (selected for the float path) is

$$Q = V_m \cdot A_m, \tag{10}$$

where $A_m$ is the average area between the upstream and the downstream sections.

Discharge measurements using the F method under favorable conditions may be accurate to within ±10%, but if a poor reach is selected and not enough float runs are made, the results can be as much as 25% in error [33].

In this study, the F method was selected because of its easy, efficient, and inexpensive use, while the more difficult, time-consuming CM method was adopted for its much higher accuracy. In this work, the accuracy of the two methods was evaluated by comparing the F method measurements to the CM method measurements, which were taken in some specific cross-sections of the investigated channel. We sought a fair compromise between measurement complexity and accuracy.

### 3.2. Measurement Equipment

The instrument used for the EC and temperature (T) measurements was a Delta OHM HD2106.2 equipped with a datalogger and internal memory, which provided EC values at a standard temperature of 20 °C or 25 °C. The thermoconductivity probe had a T measurement range of −50 °C/+90 °C (accuracy +/−0.2 °C, resolution 0.1 °C). The EC measurements covered various orders of magnitude of salinity ranging from 0.0 μS cm$^{-1}$ up to 2000 mS cm$^{-1}$ (accuracy +/−1% of full scale, resolution 1 μS cm$^{-1}$) if a proper cell constant value was set. For the measurements of the Tara Springs, a standard temperature of 20 °C was chosen, and a cell constant K = 1 cm$^{-1}$ was adopted, which allowed measurements to be made from low to relatively high EC.

For the application of the CM method, which was based on punctual speed measurements at different depths in the channel, a microreel and a hydrometric reel were used. The measurements provided by microreels can be disturbed by many materials, such as debris or filamentous algae, which can extend from the channel bottom to the surface, thus altering the rotation velocity of the rotor. For the determination of the flow rate of the Tara derivation channel, two reels with a propeller axis parallel to the current were used, specifically a laboratory microreel (a Nixon Streamflo Velocity Meter 400 model) and a hydrometric reel (an SIAP ME4001 model). The first, which was designed to measure low stream speeds in open channels, is intended primarily for use in clean waterways. The measuring head, which has a cage about 1.2 cm in diameter, allows for measurements in very small spaces. On the contrary, the ME4001 SIAP hydrometric reel can be used for small and large speeds in large- and small-scale waterways, in canals, and in conduits with clear, turbid, and salty waters.

For the application of the F method, different floating objects, including partially submerged spherical and vertical stick objects, were tested to identify the best method (to compromise between accuracy and feasibility), considering the multiple measurement sections and flow monitoring campaigns of the planned investigation. Specifically, two types of floats were used: a spherical object approximately 10 cm in diameter (e.g., the diameter of an orange), so as to be almost completely submerged, and a velocity rod consisting of a plastic tube closed at its ends and weighed down with quartz sand to guarantee a sinking of 45 cm.

## 4. Results

### 4.1. Conductivity and Temperature Measurements

Considering the investigated hydrogeological system, the monitoring of T and EC was mainly oriented toward identifying and characterizing the different spring sources present in the intake chamber, along the derivation channel, and in the Tara River. For this purpose, several measurement transects were selected along the channel, both upstream and downstream of the confluence with the Tara River (Figure 2).

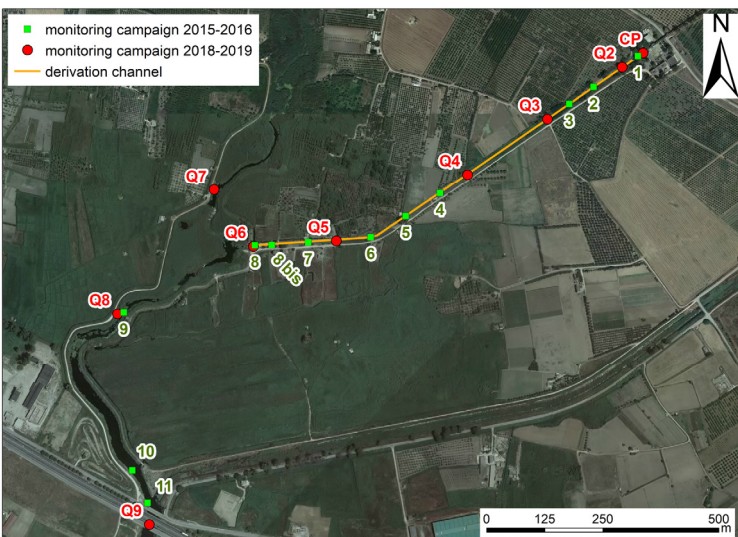

**Figure 2.** Electrical conductivity (EC) and temperature (T) measurement sections along the derivation channel and the Tara River.

In order to take into account the thermo-saline characteristics of different hydrological and climatic conditions, a total of seven monitoring campaigns were performed between January 2016 and February 2019. The position of the measurement transects shifted slightly over time due to the changing state of the chosen sections and their accessibility. To distinguish the measurement points of the 2015–2016

campaign from those of the 2018–2019 campaign, they were labeled differently (Figure 3). All of the EC and T data are shown in Figure 4. During all measurement surveys, the pumping station supplied a flow rate of about 0.500 m$^3$ s$^{-1}$, so a comparison of the data from similar conditions was possible.

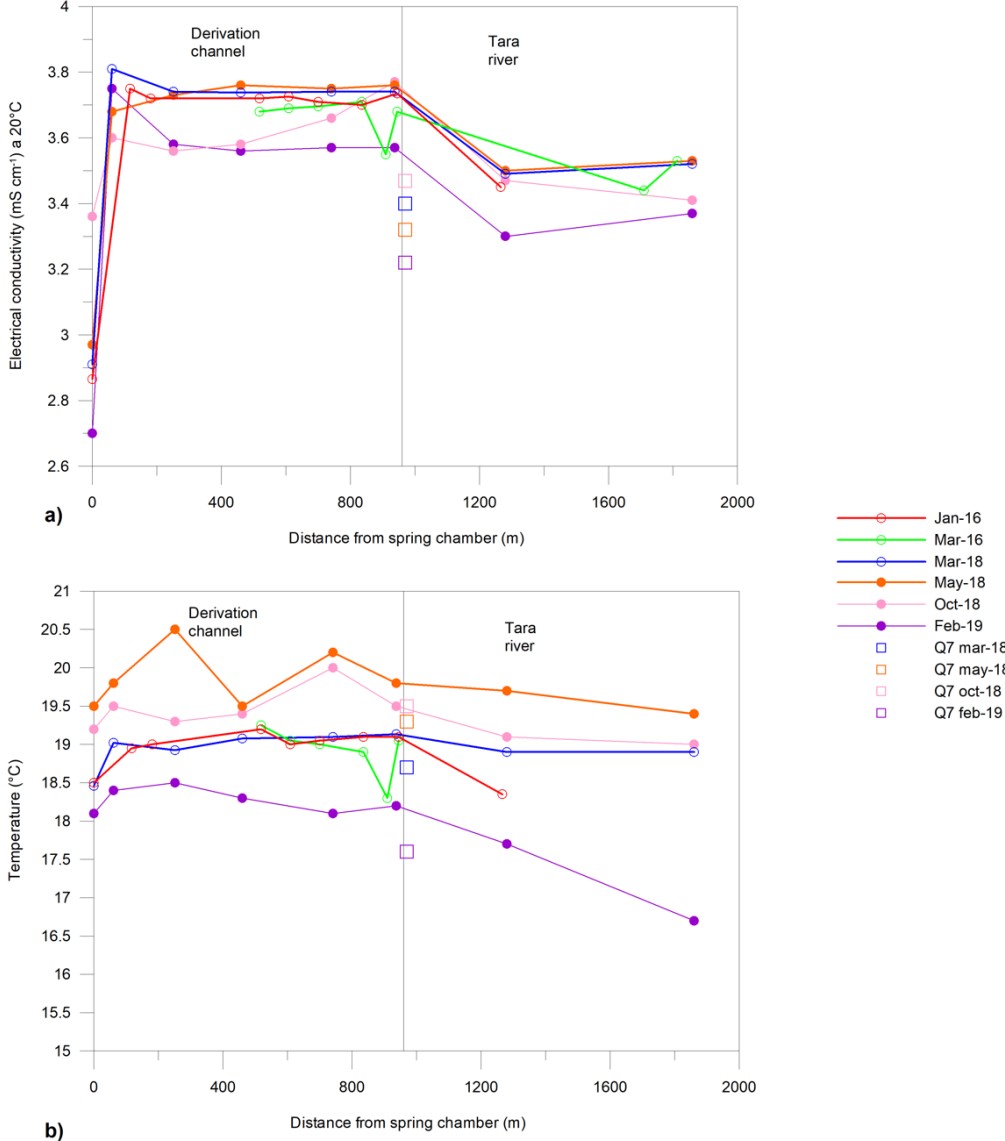

**Figure 3.** Comparison of the trends in (**a**) electrical conductivity and (**b**) temperature in the spring intake chamber, along the derivation channel, and in the Tara River during the monitoring campaigns.

Moving from the spring intake chamber to the channel, an abrupt rise in salinity could be observed. The EC values remained roughly constant in the channel, while a reduction in salinity was registered after the confluence with the Tara River and in the section Q7, which was the only section upstream of the confluence (Figure 3a).

Moreover, it was evident that EC variations over time were negligible for the same measurement section. A more evident variation could be observed for the 2019 measurements, which was an effect of abundant autumn rainfall that recharged the area of the aquifer feeding the Tara spring system.

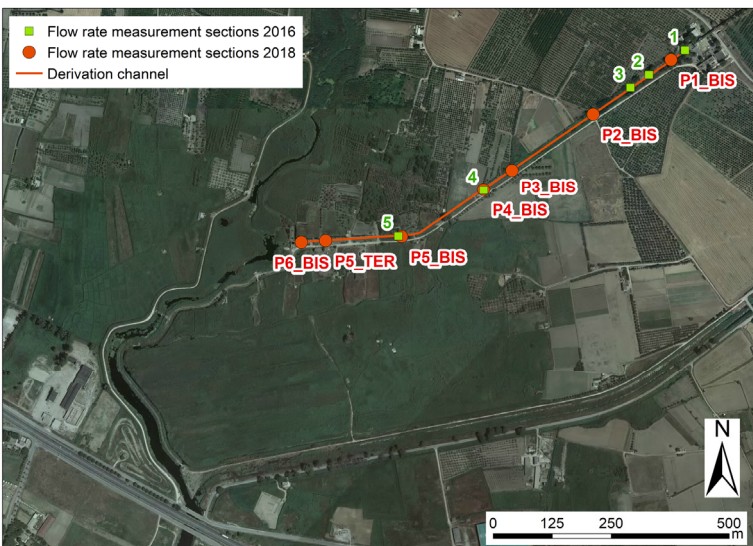

**Figure 4.** Flow rate measurement sections along the derivation channel.

The T variations along the measurement sections roughly reflected those of EC, i.e., a lower T could be noted for the spring intake chamber, as could a reduction in temperature downstream of the confluence with the Tara River (Figure 3b). Unlike EC, T was more variable over time, as it was more easily conditioned by the air temperature. Indeed, we observed a trend linked to the season during which the measurements were taken. Nevertheless, the T values of the Tara spring system remained high compared to nearby rivers with a shallow groundwater feeding system or with prevalent rainfall input.

The T and EC variations characterizing the measurement sections along the Tara spring system could be linked to the presence of distinct springs fed by circuits with different depths and travel times (which means different interaction degrees with the rock and mixing with waters with different degrees of salinity). This is in agreement with previous studies that have highlighted the presence, within the springs area, of sectors characterized by different EC values [41].

On the basis of the above remarks, it was possible to identify with some confidence the presence of additional springs between Sections 1 and 2, Sections 2 and 3, and Sections 5 and 6.

*4.2. Discharge Measurements Using the CM Method*

On the basis of the initial indications regarding the location of additional springs along the channel, which were deduced from the T and EC monitoring campaigns, the first two flow rate measurement campaigns took place on 29 January and 22 March 2016. Considering the regular lineament of the channel (a single turn about 700 m from the intake chamber), the selection of the cross-sections took into account having sufficient channel portions with a smooth bottom and banks to guarantee minimal flow turbulence and a representative hydraulic gradient. Depending on the approximately 3-m width of the stream, between 5 and 6 vertical sections were adopted for the flow measurements. At first, the geometry of each cross-section was surveyed, i.e., its width, bottom depth, and water level, moving from the right to left bank. At the same time, velocity measurements were taken along different vertical profiles. As happened with the qualitative measurements, the position of the flow rate measurement sections changed during the monitoring period. To distinguish the measurement sections of the 2018–2019 campaign from those of the 2016 campaign, a different label was used by adding the suffix "Bis" or "Ter" (Figure 4).

During the first measurement campaign (29 January 2016), the velocity was measured in five sections (from 1 to 5 in Figure 4) using a microreel Streamflow Velocity Meter 400 at 60% ($Y_{0.6}$) of the water level for each vertical investigated (Table A1, Appendix A) (as an approximation of the average

velocity). Only for Section 3 was the velocity measured along the vertical center line (i.e., $x_i = 1.5$ m) with a 0.10-m vertical step, starting from 0.05 m below the free water surface up to 1.49 m, with deeper measurements being impossible due to dense vegetation (Table A2, Appendix A). The presence of filamentous algae is one of the possible causes of the measurement error, together with the practical difficulty of operating in stationary conditions while using a hydrometric rod over 3 m in length with the propeller of a microreel fixed at one end.

The velocity data were processed using the CM method in order to obtain estimates of the flow rates in the sections under study. Calculations were made for the area of each segment in which the section was split ($Ai$) as well as for the average velocity ($V_m$) as a weighted average of the mean velocities (at $Y_{0.6}$). The flow rate ($Q$) of the section was then calculated by multiplying the average velocity by the wet surface of the section. All data are summarized in Table 2.

**Table 2.** Results using the current meters (CM) method, which was applied to data collected during the first monitoring campaign (29 January 2016).

| Section | Subarea (m$^2$) | | Wet Area (m$^2$) | $Vm$ (cm s$^{-1}$) | $Q$ (m$^3$ s$^{-1}$) |
|---------|------|-------|------|------|------|
| 1 | A1 | 0.71 | 7.39 | 3.49 | 0.258 |
|   | A2 | 1.16 | | | |
|   | A3 | 1.15 | | | |
|   | A4 | 1.15 | | | |
|   | A5 | 1.15 | | | |
|   | A6 | 1.15 | | | |
|   | A7 | 0.93 | | | |
| 2 | A1 | 1.02 | 5.86 | 4.41 | 0.258 |
|   | A2 | 1.01 | | | |
|   | A3 | 1 | | | |
|   | A4 | 0.99 | | | |
|   | A5 | 0.94 | | | |
|   | A6 | 0.91 | | | |
| 3 | A1 | 1.08 | 6.63 | 5.04 | 0.334 |
|   | A2 | 1.1 | | | |
|   | A3 | 1.12 | | | |
|   | A4 | 1.13 | | | |
|   | A5 | 1.12 | | | |
|   | A6 | 1.1 | | | |
| 4 | A1 | 1.29 | 7.04 | 6.06 | 0.427 |
|   | A2 | 1.24 | | | |
|   | A3 | 1.24 | | | |
|   | A4 | 1.29 | | | |
|   | A5 | 1.18 | | | |
|   | A6 | 0.798 | | | |
| 5 | A1 | 0.95 | 5.63 | 3.79 | 0.213 |
|   | A2 | 0.96 | | | |
|   | A3 | 0.96 | | | |
|   | A4 | 0.92 | | | |
|   | A5 | 0.89 | | | |
|   | A6 | 0.95 | | | |

Unfortunately, for the abovementioned reasons, the flow rate values calculated using this method cannot be considered to be representative of the actual flow rate of the channel. However, it was possible to correct the calculated flow rate using the data collected from the vertical centerline of Cross-section 3, as follows: First, the vertical velocity profile, shown in Figure 5, was fitted by a fifth-degree polynomial function. Then the average velocity was obtained by considering an integral mean value theorem. Using this procedure, the average velocity of the vertical centerline of Section 3 appeared to be double (and therefore more realistic) that measured at the depth $Y_{0.6}$.

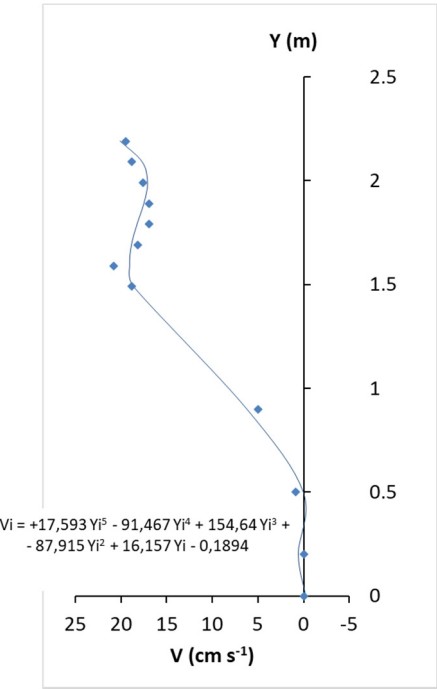

**Figure 5.** Velocity profile of the centerline in Section 3, which was obtained during the first monitoring campaign (29 January 2016).

The absolute error was calculated and, assuming this constant error for all verticals and adding it to the velocity values assessed at $Y_{0.6}$, new average velocities were calculated. On the basis of the data derived from this model, the flow rate in Section 3 was recalculated, reapplying the segments method and obtaining more realistic flow rate values (Table 3).

**Table 3.** Corrected values of velocities and flow rate for Cross-section 3, which were obtained during the first monitoring campaign (29 January 2016).

| Section | Wet Area $(m^2)$ | $Xi$ $(m)$ | $V$ at $Y_{0.6}$ $(cm\ s^{-1})$ | $V_{corrected}$ at $Y_{0.6}$ $(cm\ s^{-1})$ | $V_m$ $(cm\ s^{-1})$ | $Q$ $(m^3\ s^{-1})$ |
|---------|---------|------|------|------|------|------|
|  |  | 0.5 | 4.77 | 9.96 |  |  |
|  |  | 1 | 4.13 | 9.32 |  |  |
| 3 | 6.63 | 1.5 | 5.41 | 10.60 | 9.92 | 0.658 |
|  |  | 2 | 5.41 | 10.60 |  |  |
|  |  | 2.5 | 4.13 | 9.32 |  |  |

The velocity measurements for the second campaign (22 March 2016) were performed using a traditional hydrometric reel for river measurements, i.e., the ME 4001 model. The measurements were limited to Sections 4, 5, and 6 due to the presence of dense vegetation that made the channel

inaccessible. Moreover, in Sections 4 and 5, it was not possible to derive velocity profiles along the entire depth due to the presence of filamentous algae within the channel (Table A3, Appendix A). Therefore, we first calculated the fraction of the flow rate relative to the upper part of the section, i.e., from the free surface up to 25% of the depth ($Y_{0.25}$) for all sections. Subsequently, the lower fraction of the flow rate was calculated for the six sections. Assuming this value to be constant in all sections, it was possible to calculate the total flow in Sections 4, 5, and 6 as the sum of the two rates (Table 4). This choice was justified by the fairly regular geometry of the channel and the almost uniform distribution of the submerged vegetation.

**Table 4.** Calculated flow rate in Sections 4, 5, and 6 from the second monitoring survey (22 March 2016).

| $Q_4$ (m$^3$ s$^{-1}$) | $Q_5$ (m$^3$ s$^{-1}$) | $Q_6$ (m$^3$ s$^{-1}$) |
|---|---|---|
| 1.492 | 1.396 | 1.416 |

To confirm the results obtained from the second monitoring survey, the average velocity and the flow rate passing through the downstream section of the channel, i.e., Section 6, were calculated by adopting Kouwen et al. experimental theory [34], i.e., Equation (1) and Equation (5). The flow rate turned out to be equal to 1.439 m$^3$ s$^{-1}$, a value similar to that obtained using the segments method (Table 4).

*4.3. Discharge Measurements Using the F Method*

The 2016 flow rate measurement campaign was also done by using the simpler F method. The flow rate measurement points were the same as those used for the application of the CM method (Figure 5), excluding Section 6, which was not measured during the second monitoring campaign. At each point, the F method mandated the identification of two sections (upstream and downstream section) at a distance approximately three times the width of the channel. The geometry of all sections was reconstructed in order to calculate the average area of the liquid section. The travel times of the two types of floats (from the upstream to the downstream sections) were measured to estimate the surface speed ($V_s$). Three measurements were taken for each measurement point, and the average value was assumed to be representative of the streamflow. Subsequently, the collected $V_s$ data were processed to calculate the average sectional velocity, applying three different correction coefficients, as described in Section 3.1.2. Table 5 shows the values of $V_s$ for the two types of floats, the average velocities in each section obtained by applying the three corrective coefficients ($V_{m\ quadr}$, $V_{m\ Roche}$, $V_{mMysore}$), and finally, the transit flow rates calculated from the three speeds ($Q_{quadr}$, $Q_{Roche}$, $Q_{Mysore}$). Among these three values, $Q_{quadr}$ was chosen as the reference because it gave intermediate values compared to the other methods.

Comparing the average flow rate obtained with the two types of floats, the values were not very different from each other (±6%). This tiny difference could have been related to the presence of vegetation, which, when closer to the surface, could have hindered the free movement of the partly submerged rod. Moreover, increasing values were observed along the channel (from Section 1 to Sections 4 and 5) until values above 1.50 m$^3$ s$^{-1}$ were reached. Finally, compared to the flow rates in Sections 4 and 5 from the first and second campaigns, an increase in the flow rate of approximately 0.135 m$^3$ s$^{-1}$ was observed. This was probably due to the abundant rainfall in the catchment area in the period between February and March 2016.

**Table 5.** Values, for each of the five sections analyzed, of the surface velocity measured using the two types of floats ($V_s$), the average sectional velocity obtained using the three methods ($V_m$), and the relative transit flow rates ($Q$).

| Section | Campaign | Float | $V_s$ (cm s⁻¹) | $V_m$ (cm s⁻¹) quadr. | Roche | Mysore | $Q$ (m³ s⁻¹) quadr. | Roche | Mysore |
|---------|----------|-------|---------------|------|-------|--------|------|-------|--------|
| 1 | I | rod | 1.69 | 1.46 | 1.43 | 2.29 | 0.108 | 0.106 | 0.169 |
|   |   | orange | - | - | - | - | - | - | - |
| 2 | I | rod | 11.76 | 10.33 | 9.99 | 10.88 | 0.605 | 0.585 | 0.637 |
|   |   | orange | 10.57 | 9.01 | 8.99 | 9.87 | 0.528 | 0.526 | 0.578 |
| 3 | I | rod | 16.96 | 14.84 | 14.42 | 15.32 | 1.043 | 1.013 | 1.076 |
|   |   | orange | 16.16 | 13.76 | 13.74 | 14.63 | 0.966 | 0.965 | 1.028 |
| 4 | I | rod | 31.55 | 27.74 | 26.82 | 27.76 | 1.523 | 1.472 | 1.524 |
|   |   | orange | 35.62 | 30.37 | 30.28 | 31.23 | 1.667 | 1.662 | 1.715 |
|   | II | rod | 32.66 | 28.81 | 27.76 | 28.71 | 1.658 | 1.598 | 1.652 |
|   |    | orange | 34.11 | 29.08 | 28.99 | 29.94 | 1.674 | 1.668 | 1.723 |
| 5 | I | rod | 28.43 | 25.00 | 24.17 | 25.10 | 1.421 | 1.374 | 1.427 |
|   |   | orange | 28.94 | 24.67 | 24.60 | 25.53 | 1.403 | 1.398 | 1.451 |
|   | II | rod | 25.62 | 22.60 | 21.77 | 22.70 | 1.556 | 1.499 | 1.563 |
|   |    | orange | - | - | - | - | - | - | - |

## 4.4. Comparison between Discharge Measurements

We compared the flow rates obtained through the different measurement methods. Starting from the more reliable results, which were obtained during the second measurement campaign, the reliability of the F method with respect to the more rigorous CM method was assessed by comparing the same measurement sections (Table 6). From this comparison, it could be deduced that, regardless of the type of float used, the F method tended to overestimate the water flow rates by about 11% compared to the CM method. However, given the simplicity and rapidity of the former, a relative error of 11% may be widely acceptable compared to a more rigorous but much more time-consuming method, especially if multiple measurements are required with systematic repetitions in different periods of the hydrological year [38]. For this reason, the subsequent flow measurement campaigns that took place during 2018 were carried out using only the F method.

**Table 6.** Comparison between the flow rates measured using the CM method and float (F) method in Sections 4 and 5 of the derivation channel.

| Cross Section | $Q$ (m³ s⁻¹) Reel | Orange | Rod | Overestimation (%) Orange | Rod |
|---------------|------|--------|------|--------|------|
| 4 | 1.492 | 1.667 | 1.658 | 11.93 | 11.14 |
| 5 | 1.396 | - | 1.556 | - | 11.38 |

## 5. Discussion

The need for timely and cost-effective hydrological campaigns could be crucial for new water development projects or to evaluate the current degree of exploitation under altered hydrogeological regimes due to climate change effects and water overexploitation trends. In this study, EC and T measurements along the derivation channel were preliminarily carried out with the aim of identifying possible additional sources of springs existing at the bottom of the channel. Afterwards, flow rate measurements were performed in the same transects to characterize the hydrological regime. The flow rates measured using the F method are shown in Table 7. Both the measured and net (or natural) flow rate were obtained while taking into account (i.e., adding) the flow rates from the active pumping station at the time of measurement (listed below). Figure 6 shows the net flow rates calculated in the sections during all of the measurement campaigns.

**Table 7.** Summary of flow rate measurements obtained using the F method.

| Date | Withdrawal (m³ s⁻¹) | Section | Distance from Plant (m) | $Q$ (m³ s⁻¹) | |
|------|------|------|------|------|------|
| | | | | **Measured** | **Net** |
| 29/01/2016 | 0.500 | 1 | 20.5 | 0.108 | 0.608 |
| | | 2 | 92.0 | 0.605 | 1.105 |
| | | 3 | 140.0 | 1.043 | 1.543 |
| | | 4 | 520.0 | 1.523 | 2.023 |
| | | 5 | 720.0 | 1.421 | 1.921 |
| 22/03/2016 | 0.500 | 4 | 520.0 | 1.658 | 2.158 |
| | | 5 | 720.0 | 1.556 | 2.056 |
| 15/02/2018 | 0.460 | 1 | 20.5 | 0 | 0.460 |
| | | P1-Bis | 54.0 | 0.362 | 0.822 |
| | | P2-Bis | 244.0 | 0.810 | 1.270 |
| | | P3-Bis | 454.0 | 0.680 | 1.140 |
| | | P4-Bis | 526.0 | 1.266 | 1.726 |
| | | P5-Bis | 730.0 | 1.294 | 1.754 |
| | | P6-Bis | 934.0 | 0.694 | 1.154 |
| 19/04/2018 | 0.428 | P6-Ter | 934.0 | 0.841 | 1.269 |
| | | P5-Ter | 900.0 | 1.255 | 1.683 |
| 18/05/2018 | 0.400 | P1-Bis | 54.0 | 0.470 | 0.870 |
| | | P2-Bis | 244.0 | 1.082 | 1.482 |
| | | P5-Ter | 900.0 | 1.095 | 1.495 |
| | | P6-Ter | 934.0 | 0.775 | 1.175 |
| 08/06/2018 | 0.460 | 1 | 20.5 | 0 | 0.460 |
| | | P1-Bis | 54.0 | 0.308 | 0.768 |
| | | P2-Bis | 244.0 | 0.865 | 1.325 |
| | | P3-Bis | 454.0 | 0.899 | 1.359 |
| | | P4-Bis | 526.0 | 1,078 | 1.538 |
| | | P5-Bis | 730.0 | 1.340 | 1.800 |
| | | P5-Ter | 900.0 | 1.056 | 1.516 |
| | | P6-Ter | 934.0 | 1.147 | 1.607 |
| 04/10/2018 | 0.700 | 1 | 20.5 | −0.327 | 0.373 |
| | | P1-Bis | 54.0 | 0 | 0.700 |
| | | P2-Bis | 244.0 | 0.934 | 1.634 |
| | | P3-Bis | 454.0 | 0.665 | 1.365 |
| | | P4-Bis | 526.0 | 0.986 | 1.686 |
| | | P5-Ter | 900.0 | 0.603 | 1.303 |
| | | P6-Ter | 934.0 | 0.892 | 1.592 |

During all of the measurement sessions, the direction of the current in the channel (starting from a distance of 54 m from the pumping station) was always from the station toward the confluence with the Tara River, confirming the fact that the present pumping rates are lower than drainage by the derivation channel and the main spring located in the intake chamber.

For instance, the first values shown in the graph (Figure 6), i.e., those closest to the intake chamber (Section 1), were the outcome of the water flowing from the intercepted spring. An increase in the flow rates, which was observed around 50 m from the intake chamber (section P1-bis), confirmed the presence of a further spring intercepted by the channel, which had already been highlighted by the EC and T measurements. Its flow rate, which was calculated as the difference in flow between this section and the previous one, was, on average, equal to 0.332 m³ s⁻¹.

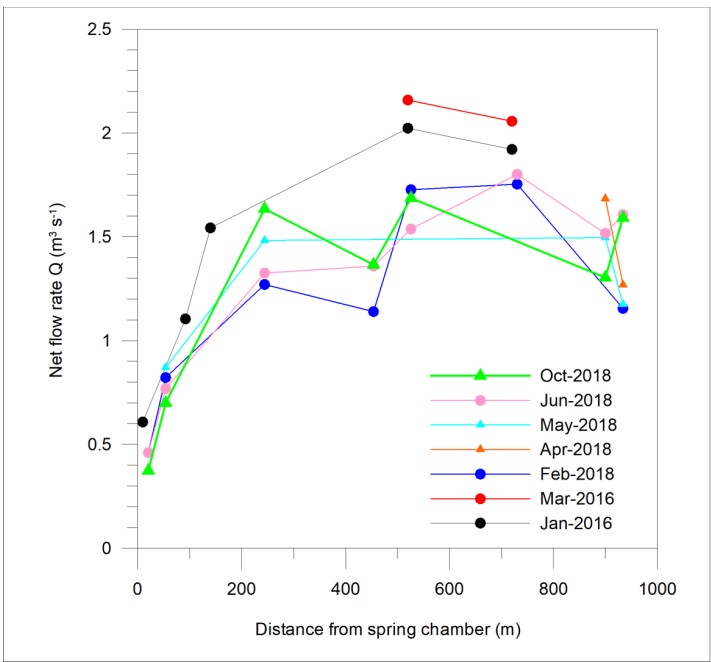

**Figure 6.** Net flow rate trend along the derivation channel.

Proceeding toward the confluence with the Tara River, around 730 m from the pumping station (Sections 5 and 5-Bis), the flow rates tended to increase up to 1.700–1.900 m$^3$ s$^{-1}$, highlighting a further spring interception along the channel with a contribution of 0.900–1.000 m$^3$ s$^{-1}$. On the other hand, in the terminal branch of the derivation channel, a reduction in flow rate was generally observed, with values of 1.100–1.600 m$^3$ s$^{-1}$, depending on the period. The observed reduction in flow rate was linked to the presence of a bypass channel with a siphon that deviated part of the channel discharge laterally. It is interesting to compare these flow measurements to those carried out in the derivation channel during the two-year period from 1980 to 1981 [42] (Table 8), a period in which the uptake from the pumping station varied between 2.400 m$^3$ s$^{-1}$ and 3.000 m$^3$ s$^{-1}$. In those withdrawal conditions, the direction of the current was always from the Tara River to the intake chamber (and is therefore indicated with negative values).

**Table 8.** Summary of flow rate measurements conducted during the years 1980–1981.

| Date | Withdrawal (m$^3$ s$^{-1}$) | Cross-Section | Distance from Plant (m) | $Q$ (m$^3$ s$^{-1}$) Measured | Net |
|---|---|---|---|---|---|
| 20/11/1980 | 2.400 | B | 66 | −1.400 | 1.000 |
| | | A | 953.5 | −0.800 | 1.600 |
| 09/02/1981 | 2.700 | 1A | 86.55 | −1.497 | 1.203 |
| | | 2A | 198.04 | −1.715 | 0.985 |
| | | 3A | 824.9 | −1.518 | 1.182 |
| | | 4A | 961.5 | −1.554 | 1.146 |
| 14/05/1981 | 3.000 | 1B | 391.95 | −1.360 | 1.640 |
| | | 2B | 595.85 | −1.520 | 1.480 |
| | | 3B | 824.9 | −1.770 | 1.230 |
| | | 4B | 916.65 | −1.660 | 1.340 |

The net flow estimations closest to the pumping station were carried out at 66 m and 86.5 m from the intake chamber (Sections B and 1A, Table 8), with values between 1.000 m$^3$ s$^{-1}$ and 1.200 m$^3$s$^{-1}$, about 10% higher than the 2016 and 2018 campaigns. Furthermore, considering the extent of the

withdrawals in 1980–1981, it can be deduced that a further 1.200–2.000 $m^3$ $s^{-1}$ was derived partly from the additional springs intercepted by the channel and partly by the Tara River. In fact, in the 1980–1981 measurements, the flow values measured at the sections near the confluence with the Tara River (indicated as A, 4A, and 4B in Table 8) express a good approximation of the amount of water derived from the natural water course (varying between 0.800 $m^3$ $s^{-1}$ and over 1.600 $m^3$ $s^{-1}$).

## 6. Conclusions

The aim of this study was to verify the residual exploitation capacity of a complex of springs fed by a large carbonate aquifer in southern Italy. The complex has been used for more than 60 years for a variety of industrial and irrigation uses. Consequently, the study provides a comparison of the advantages and disadvantages of two streamflow measurement methods in field conditions in which the repeatability of measurements in the same cross-sections was conditioned by apparently random algal blooms.

Through the integration of different monitoring techniques, the survey method adopted allowed us to identify the main source springs that feed the artificial channel. Its current water withdrawal rate is in the range between 0.500 $m^3$ $s^{-1}$ and 0.700 $m^3$ $s^{-1}$, and plans are in place to use this water to supply a future desalination plant for drinking use that will bring the total withdrawal to around 1.500 $m^3$ $s^{-1}$.

The thermo-salinity monitoring carried out at different times of the year, both along the derivation channel and downstream of the confluence with the natural watercourse, showed substantial differences due to the presence of other springs at different points of the system, which were fed by circuits characterized by different travel times and salinity. Considering the significant increase in withdrawal from the spring system, we focused on a characterization of the channel's hydrological regime, identifying the most suitable measurement method in terms of simplicity of execution and reliability of the measurements, as well as the distinctive characteristics of the channel (narrow and deep) and the observed flow regimes.

The first flow estimation sessions carried out with flow rate measurements using the CM method and the first outflow estimations using the subsections method posed considerable difficulties due to the presence of algae in the sections under consideration. The F method, although less precise (with an average error of 11%), was easier to implement and repeat over time, so it was chosen to monitor the flow in different sections of the derivation channel and during various periods of the hydrological year. The relative error between the CM and F measurements, including the development of a CM correction method based on the velocity profiles obtained in some vegetation-free cross-sections, was evaluated. Considering the flows taken from the pumping station, it was possible to estimate the natural flow rate regime, an essential element for assessing the sustainability of the pumping rate needed to supply the desalination plant. The current flow rate values from the area close to the confluence with the Tara River were determined on the basis of the measurements carried out over two years of monitoring: between 1.000 $m^3$ $s^{-1}$ and 1.500 $m^3$ $s^{-1}$. The present flow regime is different from that observed in the early 1980s, when there was a withdrawal rate between 2.400 $m^3$ $s^{-1}$ and 3.500 $m^3$ $s^{-1}$, and a conspicuous amount of water (0.800–1.600 $m^3$ $s^{-1}$) was derived from the natural water course. The final observation is that this comparison, which is reported in terms of variations in natural flow, showed a slight reduction in terms of current spring flows, probably due to the general increase in groundwater withdrawal for irrigation purposes. It can be deduced that given the considerable decrease in withdrawals at the pumping station, the portion of the spring system drained by the derivation channel is characterized by a flow rate regime, which is both consistent and stable enough to satisfy the increase in withdrawals required for the desalination plant. As an overall benefit of the adopted approach, the combined application of the implemented techniques, although not new, provides a solid, but simple and economic methodology for characterizing springs contributing flow to rivers that can be applied by local water practitioners after limited training.

**Author Contributions:** Conceptualization, I.P. and R.M.; methodology, I.P., D.M., and R.M.; formal analysis, D.M., R.M., and I.P.; data curation, I.P., R.M., M.C.C., and L.D.C.; writing—original draft preparation, D.M. and I.P.; writing—review and editing, D.M., I.P., R.M., M.C.C., and L.D.C. All authors have read and agreed to the published version of the manuscript.

**Funding:** This research was cofounded by the regional water utility Acquedotto Pugliese (AQP S.p.A.) within an investigative study commissioned for the qualitative and quantitative characterization of the Tara Springs.

**Acknowledgments:** This study was made possible following an agreement between the Regional Water Utility (AQP), the Irrigation Development Agency (EIPLI), and the Water Research Institute (IRSA-CNR). The authors express their gratitude to EIPLI for providing technical information on the Tara spring pumping system and on the local environment. The support provided by A. Fabiano, L. Ribatti, and P. Lavermicocca during the field measurement sessions and the data preparation is gratefully acknowledged.

**Conflicts of Interest:** The authors declare no conflict of interest. The funders had no role in the design of the study; in the collection, analyses, or interpretation of data; in the writing of the manuscript; or in the decision to publish the results.

## Appendix A

**Table A1.** Data from the five sections, collected during the first monitoring campaign (29 January 2016).

| Section | Distance from Spring Chamber (m) | Section Width (m) | Wet Area (m$^2$) | $X_i$ (m) | $Y_i$ (m) | Frequency at Y0.6 (Hz) | $V$ at Y0.6 (cm s$^{-1}$) |
|---|---|---|---|---|---|---|---|
| 1 | 18.5 | 3.2 | 7.39 | 0.3 | 2.35 | 1 | 3.49 |
| | | | | 0.8 | 2.3 | 1 | 3.49 |
| | | | | 1.3 | 2.3 | 1 | 3.49 |
| | | | | 1.8 | 2.3 | 1 | 3.49 |
| | | | | 2.3 | 2.29 | 1 | 3.49 |
| | | | | 2.8 | 2.32 | 1 | 3.49 |
| 2 | 118 | 3 | 5.86 | 0.5 | 2.03 | 3 | 4.77 |
| | | | | 1 | 2.01 | 3 | 4.77 |
| | | | | 1.5 | 1.99 | 2 | 4.13 |
| | | | | 2 | 1.96 | 2 | 4.13 |
| | | | | 2.5 | 1.81 | 2 | 4.13 |
| 3 | 185 | 3 | 6.63 | 0.5 | 2.16 | 3 | 4.77 |
| | | | | 1 | 2.23 | 2 | 4.13 |
| | | | | 1.5 | 2.24 | 4 | 5.41 |
| | | | | 2 | 2.27 | 4 | 5.41 |
| | | | | 2.5 | 2.19 | 2 | 4.13 |
| 4 | 520 | 2.88 | 6.86 | 0.5 | 2.58 | 4 | 5.41 |
| | | | | 1 | 2.38 | 10 | 9.24 |
| | | | | 1.5 | 2.58 | 3 | 4.77 |
| | | | | 2 | 2.6 | 2 | 4.13 |
| | | | | 2.5 | 12.1 | 7 | 7.33 |
| 5 | 710 | 3.03 | 5.63 | 0.5 | 1.9 | 1 | 3.49 |
| | | | | 1 | 1.94 | 1 | 3.49 |
| | | | | 1.5 | 1.9 | 1 | 3.49 |
| | | | | 2 | 1.76 | 4 | 5.41 |
| | | | | 2.5 | 1.8 | 1 | 3.49 |

**Table A2.** Collected data from Cross-section 3 along the vertical centerline, collected during the first monitoring campaign (29 January 2016).

| Section | $X_i$ (m) | $Y_j$ (m) | $F_j$ (Hz) | $V_j$ (cm s$^{-1}$) |
|---|---|---|---|---|
| | | 2.19 | 26 | 19.48 |
| | | 2.09 | 25 | 18.84 |
| | | 1.99 | 23 | 17.56 |
| 3 | 1.5 | 1.89 | 22 | 16.92 |
| | | 1.79 | 22 | 16.92 |
| | | 1.69 | 24 | 18.2 |
| | | 1.59 | 28 | 20.76 |
| | | 1.49 | 25 | 18.84 |

**Table A3.** Velocity vertical profiles of Sections 4, 5 and 6, which were collected during the second monitoring campaign (22 March 2016).

| | Section 4 | | | Section 5 | | | Section 6 | | | | |
|---|---|---|---|---|---|---|---|---|---|---|---|
| $X_i$ (m) | $Y_j$ (m) | $V_{ij}$ (cm s$^{-1}$) | $X_i$ (m) | $Y_j$ (m) | $V_{ij}$ (cm s$^{-1}$) | $X_i$ (m) | $Y_j$ (m) | $V_{ij}$ (cm s$^{-1}$) | $X_i$ (m) | $Y_j$ (m) | $V_{ij}$ (cm s$^{-1}$) |
| | 2.4 | 35 | | 2.05 | 19 | | 2.2 | 42 | | 2.2 | // |
| | 2.15 | 35 | | 1.8 | 19 | | 2 | 17 | | 2 | // |
| 0.5 | 1.9 | 30 | 0.5 | 1.55 | 18 | | 1.8 | 11 | | 1.8 | // |
| | 1.65 | 27 | | 1.3 | 14 | | 1.6 | 5 | | 1.6 | // |
| | 1.4 | 11 | | 1.05 | 16 | 0.5 | 1.4 | 3 | 2 | 1.4 | // |
| | 1.15 | 10 | | 0.8 | 14 | | 1.2 | 6 | | 1.2 | // |
| | 2.2 | 27 | | 0.55 | 14 | | 1 | // | | 1 | // |
| | 1.95 | 24 | | 2.09 | 16 | | 0.8 | // | | 0.8 | // |
| 1 | 1.7 | 20 | | 1.84 | 13 | | 0.6 | // | | 0.6 | // |
| | 1.45 | 10 | 1 | 1.59 | 9 | | 0.4 | // | | 0.4 | // |
| | 1.2 | // | | 1.34 | // | | 0.2 | // | | 0.2 | // |
| | 0.95 | // | | 1.09 | // | | 0.1 | // | | 0.1 | // |
| | 2.4 | 27 | | 0.84 | // | | 2.2 | 37 | | 2.2 | 22 |
| | 2.15 | 19 | | 0.59 | // | | 2 | 23 | | 2 | 22 |
| 1.5 | 1.9 | 16 | | 2.05 | 25 | | 1.8 | 21 | | 1.8 | 22 |
| | 1.65 | 9 | | 1.8 | 22 | | 1.6 | 19 | | 1.6 | 22 |
| | 1.4 | 4 | 1.5 | 1.55 | 14 | 1 | 1.4 | 19 | 2.5 | 1.4 | 18 |
| | 1.15 | // | | 1.3 | // | | 1.2 | 18 | | 1.2 | 9 |
| | 2.42 | 30 | | 1.05 | // | | 1 | 16 | | 1 | 5 |
| | 2.17 | 20 | | 0.8 | // | | 0.8 | 8 | | 0.8 | // |
| 2 | 1.92 | 18 | | 0.55 | // | | 0.6 | 8 | | 0.6 | // |
| | 1.67 | // | | 1.91 | 27 | | 0.4 | 8 | | 0.4 | // |
| | 1.42 | // | | 1.66 | 26 | | 0.2 | 9 | | 0.2 | // |
| | 1.17 | // | 2 | 1.41 | 21 | | 0.1 | 3 | | 0.1 | // |
| | 1.92 | 20 | | 1.16 | 10 | | 2.2 | 46 | | | |
| | 1.67 | 18 | | 0.91 | // | | 2 | 24 | | | |
| 2.5 | 1.42 | 9 | | 0.66 | // | | 1.8 | 20 | | | |
| | 1.17 | // | | 0.41 | // | | 1.6 | 20 | | | |
| | 0.92 | // | | 1.65 | 25 | 1.5 | 1.4 | 20 | | | |
| | 0.67 | // | | 1.4 | 22 | | 1.2 | 20 | | | |
| | | | 2.5 | 1.15 | 19 | | 1 | 17 | | | |
| | | | | 0.9 | // | | 0.8 | 13 | | | |
| | | | | 0.65 | // | | 0.6 | 11 | | | |
| | | | | 0.4 | // | | 0.4 | 8 | | | |
| | | | | 0.15 | // | | 0.2 | 5 | | | |
| | | | | | | | 0.1 | 7 | | | |

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
