# Peer review of "Combined Discharge and Thermo-Salinity Measurements for the Characterization of a Karst Spring System in Southern Italy"

_sustainability, doi:10.3390/su12083311_

Round 1
Reviewer 1 Report
Very nice compact and comprehensive article. The individual parts are logically connected, the measuring technique and procedures are sufficiently described. Attention is also paid to measurement uncertainties.
Comments:
- when writing in the text and in tables and graphs I recommend to unify the units and use the SI system and its multiples (why are used l / s, cm / s, but also right m3 / s and m / s?);
- pay attention to the texts written in Figures 1, 2, 4, 7, which are readable only at high magnification;
- it is appropriate to write the equation in Fig. 7 with respect to symbols (depth of velocity);
- line 452, tab. 8 pay attention to writing headers;
- line 465, tab. 9 incorrect CM-method - M missing;
- row 506, tab. 11 and line 104 - what is the meaning of L / s, l / s and l · s-1;
- line 507, 'Conclusion' is capter no.6;
- line 112 is the width and length of the artificial channel as the only figure for selecting the number (based on the width of the river) of the velocity measurement verticals and why the two-point velocity measurement method was chosen (the depth of measurement is correct);
- from a practical point of view, the HD2106.2 a and the Nixon Streamflo Velocity Meter 400 model are designed primarily for laboratory environments; Correction steps were chosen appropriately, as evidenced by the results achieved. However, it was possible to use not propeller anemometers, but eg ultrasonic Met-Flow, which also cover a wide speed spectrum, repeatability of measurement and correction to the temperature of flowing water.
Reviewer 2 Report
This paper is mainly describing flow measuring methods and how these can be used to extract some specific results. It has significant interest and uses a large database of field data. Some specific points on this paper are:
L40-43 the meaning of this sentence is not very clear. Also, it appears to be controversial with the general idea of your paper
L48 rephrase “the monitoring of springs involves the study of…” or something like that
L59 centuries?
L105 and later on: it would be much better to use m3 for presenting volumes.
L112 “was built as a derivation 112 channel with the aim to withdraw water through a pumping station” I guess you mean that this is a diversion channel that receives water coming from the pumping station
L115 what do you mean “either direction”
In general, L113-119 please be more clear
L132 main sources of what?
L242-244 reference
L270 use consistent units for conductivity
L478 I think you are referring to the wrong figure
Structural points
In general, the English language is in a very good level.
Abstract: Rearrange with different structure (background, knowledge gap, own method, some results). It is also suggested to not use abbreviations
In the “Study area” part I am afraid that you cannot avoid having some information about the hydrogeological system (how many aquifers, main characteristics etc)
Use consistent units throughout the whole paper.
Presenting all the data as you do it in Table 2 is not necessary. I would suggest to present a table with the final discharge for each section and put Table 2 as an annex. The same applies for Table 4.
I think Table 8 and Table 10 belong to Section 4.3, if this is not correct please ignore.
I would find interesting maybe a summary table with a comparison between the flow measurements with the CM method, the corrected CM method, the F method and corrected F method (I mean using the different coefficients). It would probably be better to have that for 1 or 2 points where you have the best data.
Scientific points
In section 3.1, although you mention this in L152, EC could be used for many of the things that you refer to but in order to have a good idea the conditions need to be very specific. This is quite rare in general. This is especially crucial to an area like yours which is close to the sea, there is some pumping (probably also in the industrial area) and the spring system is saline anyway (L104)
In the section about the CN method you are describing the technique for flow measurements in too much detail. This methodology is widely used and many of the readers will be familiar with this measuring protocol. For completion you have to mention something about the method (because you are using it) but I strongly suggest that you make this part much shorter.
In connection with the above, when performing flow measurements, the presence of vegetation is one of the attributes that need to be assessed in order to make the selection of the points where measurements will be taken. However, there are other parameters that also affect this selection (having no turns before or after the location, having no large boulders close, flow turbulence, having a representative hydraulic gradient etc). Is there a reason why you focus to the vegetation? Would it be possible to measure at a different, more suitable location?
About the F method, I would say it could be used to have an estimate of the actual discharge volumes because it is approximated using just the surface velocity value. Additionally, this method is highly affected by the turbulent flow that occurs in streams in most cases.
L332 EC changes by little, maybe 10% of the initial value, although the drop is consistent. Is this decrease significant to give an insight to the hydrological system?
It appears the in the Discussion you are extracting your outcomes mainly from comparing the flow measurements that you have taken. Since you also did the EC/T measurements too it would be nice to have a comment on how these measurements support your conclusions.
Final verdict
The paper provides us with the methods that can be used for measuring the flow in a system in order to identify changes that end up to some conclusions. The main drawback in this paper is the lack of any kind of hydrogeological description. What you are refereeing to are “valley” springs but in order to be sure you have that the water table for the area needs to be defined. Also, since this is a channel (I suppose agricultural) the changes in flow can originate from the discharge of secondary channels to the main one. Finally, in agricultural areas, pumping for irrigation is a factor that can alter the general regime by far.
The paper could change focus and be more about the methods and their application (more like a technical note) and still be interesting to the reader.
Reviewer 3 Report
- The paper is presenting the results of the flow and quality survey of a spring system in southern Italy. The paper has good quality of writing, well structured and has detailed explanation.
- Although the subject might be important for the local readers, it is not clear to me why it should be the interest of the international readers. Most of the paper presents the results of the survey using standard methods with the predictable outcome. No novel outcome which could be useful for other researcher has been presented. It is not clear to me what the benefit of this study was in terms of scientific contribution.
- For example, the flow (current) measurement methods are well-known methods and standard methods. I believe they can be removed from the text and a brief description (maximum one or two sentences) would be enough (Sections 3.1.1., 3.1.2, and 3.2)
Round 2
Reviewer 2 Report
The revised version of the paper has significant improvements in major points. The auuthors have managed to clarify many parts of the paper, upgrating the readability and the scientific content of the paper.
Reviewer 3 Report
The manuscript has been improved greatly. However, the English and editorial correction are required.
